# Anharmonic strong-coupling effects at the origin of the charge density wave in CsV$_3$Sb$_5$

Ge He [1,2,12] ✉, Leander Peis[1,3,4,10,12], Emma Frances Cuddy [5,6,12], Zhen Zhao[7], Dong Li [7], Yuhang Zhang [7], Romona Stumberger[1,3,11], Brian Moritz[6], Haitao Yang [7,8] ✉, Hongjun Gao [7,8], Thomas Peter Devereaux [5,6,9] ✉ & Rudi Hackl [1,3,4] ✉

The formation of charge density waves is a long-standing open problem, particularly in dimensions higher than one. Various observations in the vanadium antimonides discovered recently further underpin this notion. Here, we study the Kagome metal CsV$_3$Sb$_5$ using polarized inelastic light scattering and density functional theory calculations. We observe a significant gap anisotropy with $2\Delta_{max}/k_B T_{CDW} \approx 20$, far beyond the prediction of mean-field theory. The analysis of the $A_{1g}$ and $E_{2g}$ phonons, including those emerging below $T_{CDW}$, indicates strong phonon-phonon coupling, presumably mediated by a strong electron-phonon interaction. Similarly, the asymmetric Fano-type lineshape of the $A_{1g}$ amplitude mode suggests strong electron-phonon coupling below $T_{CDW}$. The large electronic gap, the enhanced anharmonic phonon-phonon coupling, and the Fano shape of the amplitude mode combined are more supportive of a strong-coupling phonon-driven charge density wave transition than of a Fermi surface instability or an exotic mechanism in CsV$_3$Sb$_5$.

Lattices of magnetic ions having regular triangular coordination are characterized by multiple ordering phenomena including ferromagnetism, frustrated antiferromagnetism, density waves and superconductivity (SC). These lattices attracted a lot of attention not only for the magnetism but also for the specific band structure being characterized by a Dirac dispersion and Weyl nodes induced by spin-orbit coupling. As a typical example, the vanadium-antimony compound class $A$V$_3$Sb$_5$ ($A$ = K, Rb, Cs) forming a Kagome lattice with alternating hexagons and triangles was discovered recently[1–4]. The V-Sb Kagome layers are separated by Sb honeycomb-like layers and alkali monolayers as shown in Fig. 1a. At low temperature, charge density waves (CDW) and SC may occur. The focus here is placed on the CDW transition forming a 2 × 2 × 2 superlattice at $T_{CDW}$ in the 100-Kelvin range which may be driven by an unconventional mechanism beyond electron–phonon interaction. Rather, the proximity to a Van Hove singularity close to the Fermi surface is considered responsible for the instability[5].

Obviously, the ordering vector **Q** connects Γ and M points (see the green arrow in Fig. 1d), in agreement with the electronic structure predicted theoretically[6] and observed by angle-resolved photoemission spectroscopy (ARPES)[7] and scanning tunnelling spectroscopy (STS)[8]. Yet, the meaning of the observed energy scales

[1]Walther Meissner Institut, Bayerische Akademie der Wissenschaften, Garching 85748, Germany. [2]Department of Physics, University College Cork, College Road, Cork T12 K8AF, Ireland. [3]School of Natural Sciences, Technische Universität München, Garching 85748, Germany. [4]IFW Dresden, Helmholtzstrasse 20, Dresden 01069, Germany. [5]Department of Materials Science and Engineering, Stanford University, Stanford, CA 94305, USA. [6]Stanford Institute for Materials and Energy Sciences, SLAC National Accelerator Laboratory and Stanford University, 2575 Sand Hill Road, Menlo Park, CA 94025, USA. [7]Beijing National Laboratory for Condensed Matter Physics, Institute of Physics, Chinese Academy of Sciences, Beijing 100190, China. [8]School of Physical Sciences, University of Chinese Academy of Sciences, Beijing 100049, China. [9]Geballe Laboratory for Advanced Materials, Stanford University, Stanford, CA 94305, USA. [10]Present address: Capgemini, Frankfurter Ring 81, 80807 München, Germany. [11]Present address: Robert Bosch GmbH, Robert-Bosch-Campus 1, 71272 Renningen, Germany. [12]These authors contributed equally: Ge He, Leander Peis, Emma Frances Cuddy. ✉e-mail: ghe@ucc.ie; htyang@iphy.ac.cn; tpd@stanford.edu; r.hackl@ifw-dresden.de

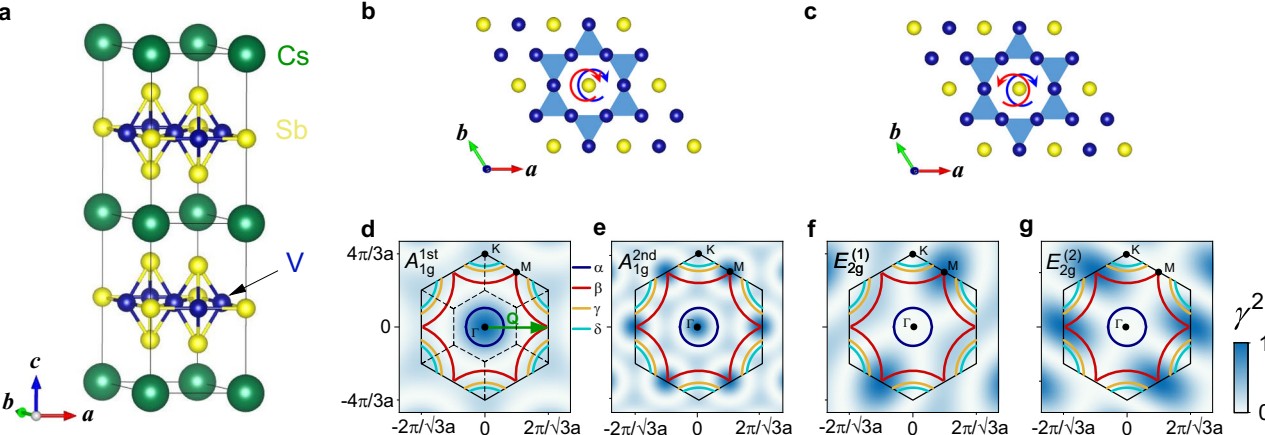

**Fig. 1 | Structure and polarization configurations in CsV₃Sb₅ with the related Raman vertices. a** The crystal structure. Cs, V, and Sb atoms are shown in green, blue and yellow, respectively. **b**, **c** Kagome lattice of the V-Sb layers. The polarization configurations of $A_{1g}$ and $E_{2g}$ symmetries are superimposed as blue and red circular arrows. The Raman vertices are shown with the colour mapping for the **d** first- and **e** second-order $A_{1g}$ symmetry, **f**, **g** first-order $E_{2g}$ symmetry. The first

Brillouin zone is represented by the black hexagon. The dark blue (electron-like pocket $\alpha$), red (hole-like pocket $\beta$), orange (hole-like pocket $\gamma$) and cyan (electron-like pocket $\delta$) curves indicate the Fermi pockets. The green arrow in (**d**) illustrates the ordering vector **Q**. The dashed lines in (**d**) indicate the corresponding folded Brillouin zone.

is still controversial. ARPES[9] and STS[10,11] find a gap at 20 meV and thus just one-fourth of the scale observed by infrared spectroscopy[12,13]. More recent ARPES measurements disclose that the small gap may originate from massive Dirac points[14], and a larger CDW gap may open at the $M$ points[14,15] and corresponds to a ratio $2\Delta/k_B T_{CDW} \approx 20$ far beyond the weak coupling prediction of 3.53. Complementary to spectroscopic methods thermodynamic studies indicate a divergence in the heat capacity being more compatible with a first-order rather than a second-order transition as usually expected for a CDW[2].

There are various experimental methods that can be used to attack this issue. One may look for anomalies close to the ordering vector **Q** in the acoustic phonon branches using either neutron[16] or inelastic X-ray scattering[5]. This search has been unsuccessful so far, and the conclusion reached is that either k-dependent electron–phonon coupling or electron-electron interaction is the origin of CDW ordering. Alternatively, optical phonons displaying renormalization effects at $T_{CDW}$[17–19] or Fano-type line shapes may indicate strong electron–phonon coupling. In addition to phonons, oscillations of both the amplitude and the phase of the order parameter are expected for a CDW system[20]. For symmetry reasons, Raman scattering and time resolved techniques project the amplitude mode (AM) directly thus tracking the CDW phase transition[18,19,21–23]. In weak-coupling systems, the AMs are expected to have a symmetric Lorentzian line-shape[21,23] with increasing width upon approaching $T_{CDW}$ from below. It is not clear which effect on the AM may be expected if the coupling increases substantially. Finally, the CDW electronic gap is accessible by light scattering.

In this paper, we address the open questions as to the states involving the formation of CDW order, including the size and momentum dependence of the electronic gap, the renormalization of phonons, and the evolution of collective modes, by investigating the temperature and polarization dependent inelastic light scattering response in CsV₃Sb₅. In particular, in contrast to the AMs found in other well-known CDW materials, we observe the $A_{1g}$ AM to be asymmetric in CsV₃Sb₅, exhibiting a strong Fano resonance. These results along with the strong anharmonic decay of the two prominent Raman-active phonons and most of the CDW-induced phonons highlight the importance of a cooperation between strong phonon–phonon and electron–phonon coupling in the formation of CDW in CsV₃Sb₅.

## Results

### Electronic continuum

Figure 2 shows the $A_{1g}$ and $E_{2g}$ Raman spectra of CsV₃Sb₅ in the range from 50 to 3600 cm⁻¹ above and below $T_{CDW}$. There is a symmetry-dependent redistribution of spectral weight from below to above the intersection points at approximately $1400 \pm 50$ and $1540 \pm 50$ cm⁻¹ in the $E_{2g}$ and the $A_{1g}$ spectra, respectively, which was not reported before. The redistribution of the spectral weight is well reproduced for different laser energies (see Supplementary Materials C for details). There is no sharp onset, rather the spectra are continuous similar to earlier observations in 2D CDW systems[24].

Upon warming the amplitude of the redistribution decreases and disappears completely above $T_{CDW}$ as shown in the insets of Fig. 2a, b, where we plot the difference between the spectra measured slightly above $T_{CDW}$ and those below. The difference spectra reveal additional features close to 600 cm⁻¹ (75 meV) and 450 cm⁻¹ (56 meV) for $A_{1g}$ and $E_{2g}$ symmetries, respectively, and suggest that the high-energy part of the $E_{2g}$ spectra consists of two distinct temperature-independent structures at $2100 \pm 200$ and $3000 \pm 200$ cm⁻¹, whereas there is only a board peak at $2500 \pm 200$ cm⁻¹ in the $A_{1g}$ spectra (large gap).

Along with the experiments we performed DFT simulations as presented in Fig. 2c. The joint density of states (see Methods and Supplementary Materials G for more details) is determined for the pristine lattice and for the two distortions (see the insets of Fig. 2c) allowed by symmetry below $T_{CDW}$. For the tri-hexagonal (iSoD) distortion the reduction in spectral weight below 2000 cm⁻¹ is bigger than for the SoD case, where the effects of the distortion can barely be seen. Thus, the electronic Raman spectra favour the same distortion as the analysis of phonon instabilities[19]. In addition, in agreement with experimental observations, large and small gap features are clearly identified. The mismatch in energy between the DFT calculations and the experiments may be reconciled by considering a renormalization factor of -1.67 to be expected for the band energies in this material class having strong electronic correlations[25].

### Phonons and amplitude modes

Two prominent Raman-active phonon lines are observed saturating at 137.5 and 119.5 cm⁻¹ in the zero-temperature limit for $A_{1g}$ and $E_{2g}$ symmetry, respectively, as shown in Fig. 3. They have previously been identified by Raman scattering[17–19,26]. Both the $A_{1g}$ and the $E_{2g}$ phonon show weak but significant renormalization effects at $T_{CDW}$ (see Fig. 3).

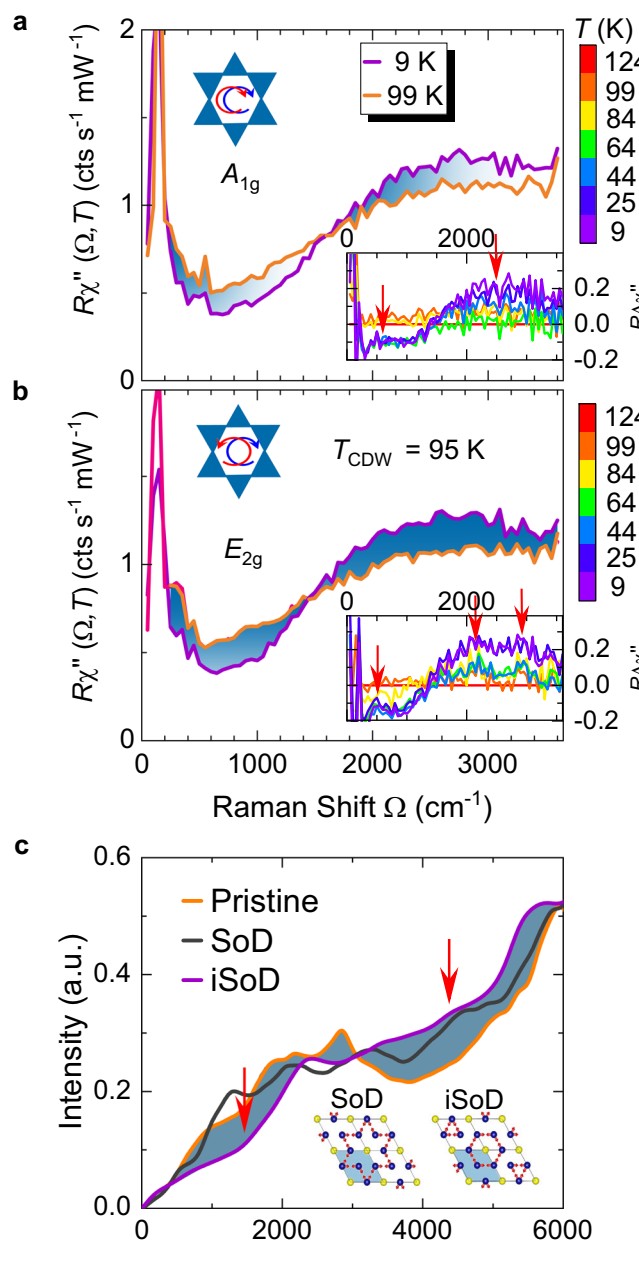

**Fig. 2 | CDW gap excitations in both $A_{1g}$ and $E_{2g}$ symmetry. a, b** Raman response above and below $T_{CDW}$ in $A_{1g}$ and $E_{2g}$ symmetry. The redistribution of the spectral weight is highlighted by the cyan areas. The insets show the difference between spectra at 124 K and the low-temperature spectra, with temperature indicated by the colour bars on the right. The small and large gap features are indicated by the red vertical arrows. **c** The FS-integrated electronic response is calculated from DFT using the pristine (orange), Star of David (SoD, black), and inverse Star of David (iSoD, violet) lattice. The loss and gain of intensity between the response in the pristine and iSoD-distorted lattice are highlighted by cyan areas. In agreement with the experimental results, the small and large gaps are also reproduced by DFT calculations (marked by the red arrows). Insets: The SoD and iSoD distortion in the V-Sb layer. The blue shaded area shows the pristine unit cell. The unit cell below $T_{CDW}$ is twice as large.

Upon cooling, the $A_{1g}$ phonon changes discontinuously near $T_{CDW}$ and saturates below (see Fig. 3b). Although the energies observed upon cooling and heating do not exactly coincide the hysteresis cannot be considered significant enough to support a first-order phase transition as suggested by the thermodynamic data[27]. The energy of the $E_{2g}$ line

does not exhibit significant changes across $T_{CDW}$ (see Fig. 3f). A weak dip exactly at $T_{CDW}$ may exist but our resolution is not sufficient here. In either symmetry, the line widths exhibit kinks at $T_{CDW}$ and decrease faster below $T_{CDW}$ than above (see Fig. 3c, g). The line widths are well described in terms of symmetric anharmonic decay (full lines in Fig. 3c, g)[28]. The resulting phonon–phonon coupling constants $\lambda_{ph\text{-}ph}$ are substantially enhanced below $T_{CDW}$.

Along with the measured phonon energies we show their variation with temperature expected from the volume contraction according to Grüneisen theory[29] (full lines in Fig. 3b, f) using the thermal expansion data of ref. 27 (more details can be found in Supplementary Materials E). The Grüneisen parameters $\gamma_i$ for the $A_{1g}$ and the $E_{2g}$ phonon are found to be 2.45 and 1.65, respectively, close to the typical value of 2. The expansion data confirm that the transition at $T_{CDW}$ is weakly first order since the volume is not constant across $T_{CDW}$. The anomaly of the expansion coefficient $\alpha_V(T)$ is substantial (Fig. 3h), but the volume expansion is small[27] and the effect on the phonon energies, for which $\alpha_V(T)$ is used, is even smaller. While describing the data well for $T > T_{CDW}$, the volume change cannot explain the hardening of approximately 1.2 cm$^{-1}$ of the $A_{1g}$ phonon at $T_{CDW}$. Rather, it predicts a small softening of the phonon frequency. On the other hand, the hardening of the $A_{1g}$ phonon is properly predicted by our DFT simulations (Details can be found in Table II of Supplementary Materials G). Zooming in on the region around $T_{CDW}$, a precursor of the phase transition is resolved in a range of approximately 10 K above $T_{CDW}$ (see Fig. 3d).

The weak additional lines observed below $T_{CDW}$ are indicated by black asterisks and orange diamonds in Fig. 4a and b. In the zero-temperature limit the three $A_{1g}$ lines are located at 43.0, 105.4 and 200.0 cm$^{-1}$. The six $E_{2g}$ lines appear at 43.2, 60.0, 101.0, 180.0, 208.2 and 224.0 cm$^{-1}$. The lines at 43.0 and 105.4 cm$^{-1}$ are also found in pump-probe experiments[30,31]. These emerging lines are observed at nearly the same energies for different laser excitations (see Supplementary Materials D for details). Details of the temperature-dependent positions and widths of these lines can be found in Supplementary Materials E.

The lines marked by asterisks have weak and conventional temperature dependences and soften by less than 2% between the low-temperature limit and $T_{CDW}$ (see Fig. 4c, d). The lines at 105 and 208 cm$^{-1}$ labelled with orange diamonds shift to lower energies by 17 and 10 cm$^{-1}$, respectively, upon approaching $T_{CDW}$, corresponding approximately to 15% and 5% relative shift. These two lines are identified as CDW AMs in CsV$_3$Sb$_5$. The $A_{1g}$ line broadens by approximately an order of magnitude close to $T_{CDW}$ and assumes a rather asymmetric shape in the range 40–80 K (see Fig. 5a).

## Discussion
In the following electronic excitations, phonon anomalies and collective amplitude modes (AMs) below the ordering temperature $T_{CDW}$ will be discussed.

### Excitations across the gap
The redistribution of the spectral weight below $T_{CDW}$ displaying slightly different intersection points and peak positions in the electronic $A_{1g}$ and $E_{2g}$ spectra (see Fig. 2) indicate anisotropies of the energy gap. The resistivity shows that the system remains metallic below $T_{CDW}$ and that the gap vanishes on extended parts of the Fermi surface. In contrast to a superconductor, where the gap opens symmetrical with respect to the Fermi energy $E_F$ at $\pm \mathbf{k}$, single- and two-particle spectroscopies cannot readily be compared in a CDW system. While both Raman and IR spectroscopy measure occupied and unoccupied states, thus the joint density of states with specific weighting factors, ARPES probes only the occupied states and STS either tunnels into unoccupied states or extracts electrons from occupied bands. This fact manifests itself directly in the asymmetry of the STS spectra

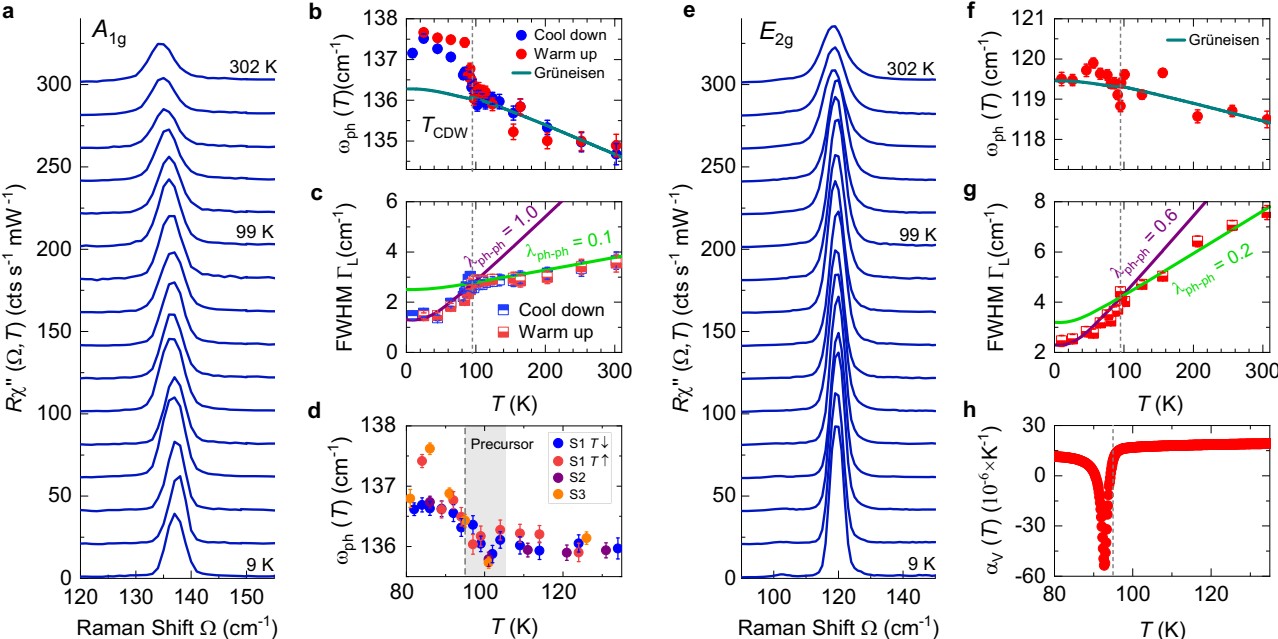

**Fig. 3 | Phonon renormalization at $T_{CDW}$. a, e** The temperature evolution of the $A_{1g}$ and the $E_{2g}$ phonon lines was measured at 9, 25, 45, 55, 65, 75, 80, 85, 89, 95, 99, 124, 154, 203, 253, 302 K, respectively. For clarity, the spectra measured above 9 K are consecutively offset by 15 cts/(s mW). **b, c, f, g** Phonon energies $\omega_{ph}$ widths (FWHM) of the two lines are derived from Voigt fits (see "Methods"). **b** A hysteresis may exist below $T_{CDW}$. The green line is derived from the volume expansion[27] using Grüneisen theory with $\gamma = 2.45$. It is adjusted to the blue data points. **d** Zoom in on the range near $T_{CDW}$ of the energy (**b**). The shaded area indicates a range of 10 K, where the $A_{1g}$ phonon exhibits a dip and an increase of $\omega_{ph}$ above $T_{CDW}$. **f** The temperature

dependence of the energy in $E_{2g}$ symmetry is weaker than for the $A_{1g}$ mode in (**b**), and the related Grüneisen parameter is thus smaller, $\gamma = 1.65$. There may be an anomaly directly at $T_{CDW}$. **c, g** Temperature dependences of the phonon line widths ($\Gamma_L$) of the $A_{1g}$ and the $E_{2g}$ phonons. The data were fitted separately below and above $T_{CDW}$ using an anharmonic model[28] (see Supplementary Materials E for details). There are obvious slope changes at $T_{CDW}$ for both the $A_{1g}$ and the $E_{2g}$ phonons. **h** Volume expansion coefficient $\alpha_V(T)$. The error bars include the statistical and systematic errors from the fitting of the phonon lines and the reproducibility of the spectrometer.

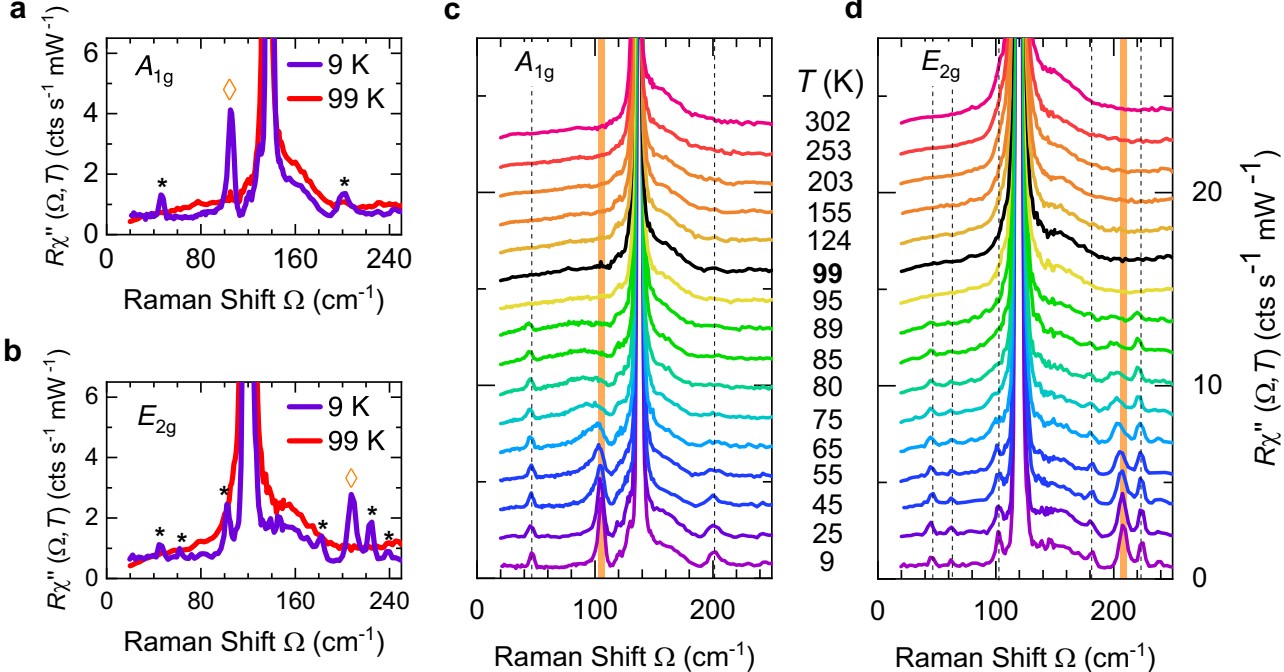

**Fig. 4 | Zone folded phonons and amplitude mode in CsV$_3$Sb$_5$. a, b** Raman spectra of CsV$_3$Sb$_5$ below and above $T_{CDW}$ in $A_{1g}$ and $E_{2g}$ symmetry. Below $T_{CDW}$, several additional peaks appear which are marked by black asterisks for the zone folded phonon lines and orange diamonds for the amplitude modes.

**c, d** Temperature-dependent Raman spectra of CsV$_3$Sb$_5$ in $A_{1g}$ and $E_{2g}$ symmetry, respectively. For clarity, the spectra are consecutively offset by 0.5 cts/(s mW) each except for those measured at 9 K.

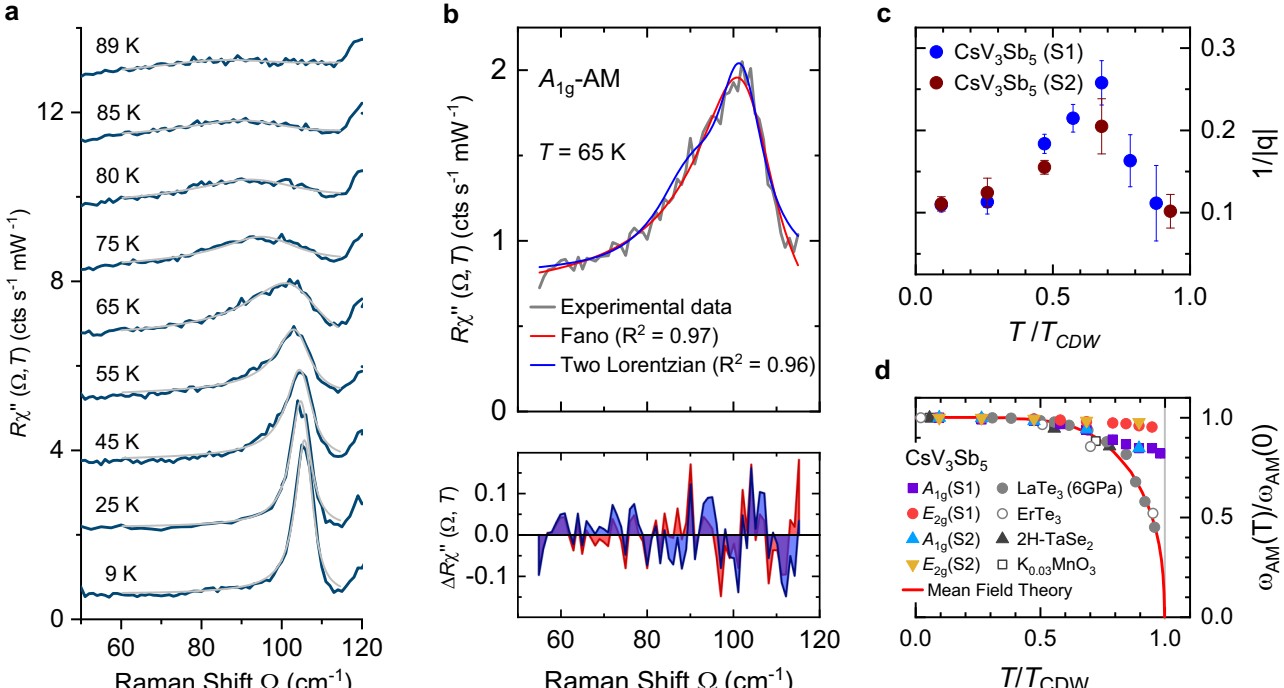

**Fig. 5 | Asymmetry of the $A_{1g}$ amplitude mode. a** Temperature evolution of the $A_{1g}$ AM. Except for the spectrum at 9 K, the spectra have been consecutively shifted by 1.5 cts/(s mW) for clarity. The Fano fits are superimposed in the spectra (light grey lines). **b** Top panel: The comparison between a superposition of two Lorentzian lines as suggested in ref. 18 and a Fano line shape to the $A_{1g}$ AM. The Fano line yields a better residuum $R^2$. Results for the entire temperature range are presented in Supplementary Materials H. Bottom panel: The deviation of the spectra from the fitting curves (Red line: Fano, blue line: two Lorentzians). **c** The asymmetry parameter $1/|q|$ as a function of normalized temperature in $CsV_3Sb_5$. The error bars are estimated from the fitting of the AM lines. **d** The temperature dependence of AM energies in various CDW materials. The data points in grey and black were extracted from refs. 23,24,40,41,50. The peak energies of both AMs obviously deviate from the prediction of mean field theory (red curve) in $CsV_3Sb_5$. S1 and S2 in (**c**) and (**d**) denote two different samples.

for negative and, respectively, positive bias[8]. Yet, in all spectroscopies, two distinct features are observed at low and high energies. For two-particle techniques, the related features are expected at approximately twice the energy observed by single-particle methods.

In systems having strongly momentum-dependent gaps, the gap energy is closer to the intersection points than the peak energies of the Raman spectra. The resulting magnitude of the larger gap $\Delta_>$ is then at approximately 88–96 meV yielding a gap ratio of $2\Delta_>/k_B T_{CDW}$ in the range 21–23. Presumably due to the different projections of IR and Raman the results of the optical conductivity peak at an energy slightly below those of Raman spectroscopy[13]. As opposed to the IR results, we do not observe significant temperature dependences of neither the intersection points nor the peak energies.

In ARPES, gap features are extracted from the symmetrized spectra of $CsV_3Sb_5$ and $KV_3Sb_5$[7,14,15]. Large gaps are found on the smaller FSs around the $K$ points of the BZ ($\gamma$ and $\delta$ bands in the notation of ref. 15) with the maxima reaching $\Delta_> \approx 80$ meV close to the $M$ points. It is also reproduced by our DFT calculations at M point (see Supplementary Materials G for details). These gaps are slightly smaller than those derived from the Raman spectra of $CsV_3Sb_5$. One possible reason are the final states into which the electrons are scattered. In ARPES electrons propagate into the vacuum whereas in Raman they are scattered into unoccupied states above $E_F$. The selection rules are likely at the origin of the differences between $A_{1g}$ and $E_{2g}$.

In addition to the large gaps, there are also small gaps around the $M$ points having energies close to $\Delta_< \approx 20$ meV[15]. The small gaps open along the $\Gamma$-$K$ line in our DFT simulation (see Supplementary Materials G for details). We identify the Raman peaks at approximately 500 cm$^{-1}$ with the smaller gaps although the single-particle gaps derived from Raman scattering, $\Delta_< \approx 30$ meV, are still too large. As a matter of fact, comparing energy scales in a CDW systems is not as

straightforward as in a superconductor because of missing the particle-hole symmetry. On the other hand, one may learn a lot on the unoccupied states after a successful implementation of the selection rules in both Raman and IR. Yet, this endeavor requires an in-depth understanding of interband scattering which is still missing[32].

**Persistent phonons**

The two phonon lines of the pristine structure persist in the distorted phase below $T_{CDW}$. There are changes at the transition in both the positions and the widths of the lines. The widths are described successfully in terms of anharmonic decay into two acoustic modes having momentum $\pm\mathbf{k}$[28] as shown in Fig. 3c, f. There is a remarkable change of slope at $T_{CDW}$ which goes along with an increase of the phonon–phonon coupling $\lambda_{ph-ph}$ by a factor of ten and three for the $A_{1g}$ and the $E_{2g}$ line, respectively. The enhanced coupling constants in the CDW phase suggest that the ph–ph coupling is not directly mediated by anharmonicity but by electron–phonon interaction which increases upon approaching $T_{CDW}$ when the gaps close.

Above $T_{CDW}$, the modes' energies are well described by the volume change using Grüneisen theory. Below $T_{CDW}$, only the $E_{2g}$ line obeys this prediction while the energy of the $A_{1g}$ phonon changes by 1.2 cm$^{-1}$ when the zero-temperature values are compared. The energy change of the $A_{1g}$ mode is a result of the new positions of the V atoms below $T_{CDW}$. It was shown that only the iSoD distortion removes the negative energies of the acoustic phonons[19]. Along with this DFT result and our own simulations for the electronic Raman spectra we conclude that the iSoD distortion is more likely. The phonon energies derived from our DFT simulations (see Supplementary Materials G, Table II) yield the same shift for both distortion patterns. On the basis of symmetry arguments we cannot distinguish between SoD and iSoD either since both belong to the same symmetry group. Similarly, we are

not in a position to comment on possible stacking sequences along the $c$-axis as suggested by X-ray studies[33,34].

## New phonon lines below $T_{CDW}$

The additional phonon lines below $T_{CDW}$ result from the lowering of the lattice symmetry as spelled out by Wu et al.[18] and Liu et al.[19]. The lattice distortion folds the phonon dispersion by a wave vector **Q** that links the $\Gamma$ and $M$ points as seen in Fig. 1d. Here, the phonons at the zone boundary are folded to the $\Gamma$ point and become Raman active[35,36]. If one considers a SoD or an iSoD distortion (see the insets of Fig. 2c) expected for $CsV_3Sb_5$ (refs. 5,11,37), the V atoms move from the $3g$ (1/2, 0, 1/2) to $12q$ $(x, y, 1/2)$ Wykoff positions, and one expects eight additional Raman-active modes (two in $A_{1g}$, four in $E_{2g}$ and two in $E_{1g}$). This figure matches the number of the new phonons in our measurements (asterisks only). Furthermore, most of the zone-folded modes quantitatively match the frequencies obtained in DFT simulations when considering the iSoD distortion[19]. The new lines have similarly large phonon–phonon coupling constants $\lambda_{ph-ph}$ as the strong lines appearing above and below $T_{CDW}$ (see Fig. 3d, f and Table I in the Supplementary Materials E), indicating strong phonon–phonon coupling.

## Amplitude modes

The lines at 105 and 208 $cm^{-1}$ in $A_{1g}$ and $E_{2g}$ symmetry, respectively, have significantly stronger temperature dependences than the other lines appearing below $T_{CDW}$ and are identified as AMs. Yet, the variation is much weaker than predicted by mean-field theory (see Fig. 5d) and observed for the tritellurides, e.g., refs. 23,24. There may be various reasons for the deviations: (i) Impurities lead to a saturation of the AM frequency at approximately the impurity scattering rate[38]. Here, this would imply a rather disordered system with an electronic mean free path of only a few lattice constants. (ii) An effect of strong electron–phonon coupling seems more likely, although enhanced coupling does not necessarily entail a deviation from mean-field theory. Since Ginsburg-Landau theory[18] is applicable only close to the transition, where no data are available, the study of an extended temperature range below $T_{CDW}$ may be deceptive. In addition, the AM is not directly related to the gap, where single-particle (STS, ARPES) and two-particle (IR, Raman) results may return significantly different results, but rather to a soft mode above $T_{CDW}$. (iii) Strong phonon–phonon coupling and, consequently, higher order contributions from the phonons are not unlikely since the coupling $\lambda_{ph-ph}$ of all modes below $T_{CDW}$ is substantial (see Fig. 3c, g as well as Table I in Supplementary Materials E). This effect is predicted to enhance $2\Delta/k_BT_{CDW}$ substantially and induce deviations from the mean-field temperature dependence of the AMs[39]. As mentioned above the enhanced phonon–phonon coupling is most likely mediated by a substantial electron–phonon coupling entailing the asymmetry of the $A_{1g}$ amplitude mode.

The anomalies of the $A_{1g}$ line at 105 $cm^{-1}$ are incompatible with conventional phonons. Right below $T_{CDW}$ the line width is as large as 50 $cm^{-1}$ (more details can be found in Supplementary Materials E). Previously the asymmetry has been interpreted in terms of two superimposed lines having individual temperature dependences[18] or a hybridisation with CDW-induced lines[19]. We did not observe a double structure at low temperature for any of the three excitation energies studied (see Supplementary Materials D) but rather a narrow, yet asymmetric, line having a width (FWHM) of approximately 6 $cm^{-1}$ at 8 K. We tested both hypotheses and found the Fano line to reproduce the data better in the entire temperature range (see Fig. 5b). For describing the mode we used the simplified Fano formula where the line width $\Gamma$ is much smaller than the resonance energy $\omega_{AM}$, and $1/|q|$ is the asymmetry parameter,

$$I(\omega) = \frac{I_0}{|q^2+1|} \frac{(q+\varepsilon)^2}{1+\varepsilon^2}; \quad \varepsilon = 2\frac{\omega-\omega_{AM}}{\Gamma}. \quad (1)$$

The description in terms of a Fano shape yields monotonous temperature dependences of both width and resonance energy whereas the superposition of two Lorentzian lines yields erratic temperature variations as shown in Supplementary Materials H thus favoring a Fano resonance as the origin of the asymmetric amplitude mode. $1/|q|$ becomes maximal at 68 K where the transition from $2 \times 2 \times 2$ to $2 \times 2 \times 4$ stacking is observed by X-ray diffraction[33] (see Fig. 5c). The decrease of $1/|q|$ towards zero temperature is a result of the opening of the CDW gap below 1500 $cm^{-1}$ which reduces the continuum (see Fig. 2a). The Fano shape of the AM is unique in $CsV_3Sb_5$ and has not been observed in other well-known CDW materials, such as $ErTe_3$[24], $LaTe_3$[23], $2H$-$TaSe_2$[40] or $K_{0.3}MoO_3$[41], where all AMs have a symmetric Lorentzian line-shape (See Supplementary Materials I).

The asymmetric AM, along with the missing soft mode behaviour in the acoustic branches[5,16], the large $2\Delta/k_BT_{CDW}$ ratio, and the weakly first-order phase transition argue against the weak-coupling picture. In addition, we derive signatures of strong phonon–phonon coupling from the anharmonic decay of the majority of the Raman-active optical phonons (see Fig. 3c, g and Supplementary Material E), proposed by Varma and Simons as an important ingredient for strong coupling[39]. These observations supplement earlier work and highlight the interrelation of various interactions conspiring to drive the CDW.

Usually one argues that strong fluctuations suppress $T_{CDW}$ in systems having a large gap. In some materials such as $ErTe_3$ electronic fluctuations can directly be observed[24] above $T_{CDW}$. $ErTe_3$ is in fact a very clean compound and may therefore be considered a textbook example. Yet, it seems unlikely that impurities alone can explain the absence of fluctuations in $CsV_3Sb_5$ (see Supplementary Material F for details). The phonon anomalies close to $T_{CDW}$ (Fig. 3d) could indicate a narrow fluctuation regime similar to the one- or two-Kelvin range above the magnetic transition in $MnSi$[42].

In summary, we performed a polarization- and temperature-dependent Raman scattering study of the Kagome metal $CsV_3Sb_5$. The electronic continua in both the $A_{1g}$ and $E_{2g}$ symmetry exhibit a spectral-weight redistribution below the charge-density-wave transition temperature, $T_{CDW} \approx 95$ K. This redistribution indicates an energy gap of $2\Delta \lesssim 1500$ $cm^{-1}$ (185 meV) corresponding to $2\Delta/k_BT_{CDW}$ close to 22. In addition, we observe features in the range 500 $cm^{-1}$ (60 meV). These magnitudes correspond qualitatively to the distinct single particle gaps $\Delta_< \approx 25$ meV and $\Delta_> \approx 80$ meV observed by ARPES[14,15]. The DFT calculations reproduce the spectral weight redistribution favouring an iSoD distortion for $T \rightarrow 0$. In the low-energy part of the spectra several phonons pop out below $T_{CDW}$ in addition to the two modes in $A_{1g}$ and $E_{2g}$ symmetry Raman active at all temperatures. The additional lines are related to the lattice distortion due to the CDW transition. Intriguingly, we identified two CDW amplitude modes having energies of $\omega_{AM}^{A1g} = 105$ $cm^{-1}$ and $\omega_{AM}^{E2g} = 208$ $cm^{-1}$ in the low-temperature limit. The $A_{1g}$ AM couples strongly to a continuum as indicated by the Fano-type line shape displaying the strongest asymmetry at the putative crossover temperature of ~60 K between $2 \times 2 \times 2$ to $2 \times 2 \times 4$ ordering[33]. The mode's temperature dependence is weaker than predicted by mean field theory. This discrepancy may result from either impurities[38] or strong coupling[39]. Since the crystals are well-ordered we consider the strong-coupling scenario including anharmonic phonon–phonon and electron–phonon coupling[39] more likely. This interpretation is consistent with the large electronic gap and the asymmetric AM. Thus, the cooperation of mode-specific electron–phonon and intermediately strong phonon–phonon coupling may be more likely a route to the CDW transition in $CsV_3Sb_5$ than, e.g., nesting.

## Methods

### Samples

Single crystals of $CsV_3Sb_5$ were grown from liquid Cs (purity 99.98%), V powder (purity 99.9%) and Sb shot (purity 99.999%) via a modified self-flux method[43]. The mixture was put into an alumina crucible and sealed

in a quartz ampoule under argon atmosphere. The mixture was heated at 600 °C for 24 h and soaked at 1000 °C for 24 h, and subsequently cooled at 2 °C/h. Finally, the single crystals were separated from the flux by an exfoliation method. Apart from sealing and heat treatment procedures, all other preparation procedures were carried out in an argon-filled glove box. The crystals have a hexagonal morphology with a typical size of $2 \times 2 \times 1 \, mm^3$ and are stable in the air. The sample used for the Raman experiments has a $T_{CDW}$ of 95 K, characterized by resistivity and in-plane magnetic susceptibility (see Supplementary Materials A for details).

## Light scattering

The inelastic light scattering experiments were preformed in pseudo-backscattering geometry. The samples were mounted on the cold finger of a ⁴He flow cryostat immediately after cleaving. For excitation, a solid-state and an Ar⁺ laser emitting at 575, 514, and 476 nm were used. In the experiments, the laser power was adjusted to maintain an absorbed power of $P_{abs} = 4.0$ mW, resulting in a heating rate of 0.5–1 K/mW. The inelastic spectra were divided by the Bose factor yielding $R\chi''$ $(\Omega, T) = \pi\{1+n(\Omega, T)\}^{-1}S(q \approx 0, \Omega)$ where $\chi''$ is the imaginary part of Raman response function, $R$ is an experimental constant, and $S(q \approx 0, \Omega)$ is the dynamical structure factor[44]. Typical phonon lines are described by Lorentzians. If the width is close to the spectral resolution or below a Voigt function (convolution of a Lorentzian and a Gaussian, where the Gaussian width is set at 4.3 cm⁻¹) has to be used.

For projecting the $A_{1g}$ and $E_{2g}$ symmetries $RR$ and $RL$ polarization configurations were used, respectively. In terms of perpendicular linear polarizations $x$ and $y$, $R$ and $L$ are given by $R = \frac{1}{\sqrt{2}}(x + iy)$ and $L = \frac{1}{\sqrt{2}}(x - iy)$, respectively. The configurations with respect to the Kagome plane are shown in Fig. 1b, c. For electronic Raman scattering the form factors are important and highlight parts of the Brillouin zone. The form factors or Raman vertices may be expressed in terms of the band curvature or crystal harmonics[44]. The first- and second-order crystal harmonics of $A_{1g}$ symmetry and the first-order crystal harmonics of $E_{2g}$ symmetry and the position of the Fermi pockets of $CsV_3Sb_5$ are shown in Fig. 1d–f and illustrate the sensitivity of the experiment (The vertices derived from the crystal harmonics can be found in Supplementary Materials B. For details see ref. 45).

## DFT simulations

DFT calculations were performed using VASP[46] with plane wave augmented (PAW) pseudopotentials and a 300 eV energy cutoff. In all calculations, the pristine, SoD, and iSoD states were considered independently as $2 \times 2$ distortions. Lattice constants were calculated with pristine structures and kept fixed in CDW states. Minimum energy CDW states were found around 1.5% lattice distortion from the pristine structure. Structural and electronic calculations were performed on each of the three states. A $17 \times 17 \times 9$ **k**-point grid was used for electronic calculations. The electronic response in the main text was approximated using the joint density of states with the following equation:

$$\chi''_{\mu\nu}(\Omega) = \sum_k \gamma^\mu_{\mathbf{k}} \gamma^\nu_{\mathbf{k}} \int d\omega A^\mu(\mathbf{k},\omega) A^\nu(\mathbf{k},\omega + \Omega) \times [f(\omega) - f(\omega + \Omega)], \quad (2)$$

where

$$A^\mu(\mathbf{k},\omega) = \frac{1}{\pi} \frac{\Gamma}{\Gamma^2 + (\omega - \epsilon^\mu_{\mathbf{k}})^2}. \quad (3)$$

In this equation, $\mu$ and $\nu$ are band indices, $\gamma^\mu_{\mathbf{k}}$ is the Raman vertex at momentum **k** and band $\mu$, $A(\mathbf{k}, \omega)$ is the spectral weight at momentum **k** and energy $\omega$, $\Omega$ is the Raman shift energy, $f(\omega)$ is the Fermi-Dirac function, $\epsilon^\mu_{\mathbf{k}}$ is the band dispersion, $\Gamma$ is the energy broadening. A

broadening of 0.02 eV was used in the results in the main text. In the calculations, the Raman vertex $\gamma^\mu_{\mathbf{k}}$ is fixed at a value of 1. Thus, selection rules are ignored for the time being.

Phonon calculations were performed using the Phonopy code package[47,48]. A $3 \times 3 \times 4$ $k$-mesh was utilized for the calculations. Our primary focus here is on the energy of the $A_{1g}$ phonon. In the pristine state, we observed a good agreement between the calculated phonons and those observed experimentally. Based on the success in previous studies[49], for calculations comparing the pristine and CDW states, we employed DFT-3 to stabilize the phonon frequencies in the distorted phases, resulting in an overall frequency shift towards higher energy while maintaining reliable relative positions. Electron–phonon coupling to the $A_{1g}$ and $E_{2g}$ modes was calculated using the frozen phonon method and found to be negligible.

## Reporting summary

Further information on research design is available in the Nature Portfolio Reporting Summary linked to this article.

## Data availability

All relevant data that support the findings of this study are presented in the manuscript and supplementary information file. All data are available upon reasonable request from the corresponding authors.

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

## Acknowledgements

We thank R.-Z. Huang and X.-L. Dong for fruitful discussions and F. Hardy and C. Meingast for providing the thermal expansion data. This work is supported by the Deutsche Forschungsgemeinschaft (DFG) through the coordinated programme TRR80 (Projekt-ID 107745057) and projects HA2071/12-1 and -3. L.P. and R.H. were partially supported by the Bavaria-California Technology Center (BaCaTeC) under grant number A3 [2022-2]. G.H. would like to thank the Alexander von Humboldt Foundation for a research fellowship. The work in China was supported by grants from the National Key Research and Development Projects of China (2022YFA1204104), the Chinese Academy of Sciences (ZDBS-SSW-WHC001 and XDB33030000), and the National Natural Science Foundation of China (61888102). Theory work at SLAC National Accelerator (E.F.C. and T.P.D.) was supported by the U.S. Department of Energy, Office of Basic Energy Sciences, Division of Materials Sciences and Engineering, under Contract No. DE-AC02-76SF00515. The computational work utilized the resources of the National Energy Research Scientific Computing Center (NERSC) supported by the U.S. Department of Energy, Office of Science, under Contract No. DE-AC02-05CH11231.

## Author contributions

G.H. and R.H. conceived the project. G.H., L.P., D.L. and R.S. performed the Raman measurements. G.H., L.P., E.F.C., T.P.D. and R.H. analysed the Raman data. Z.Z., H.-T.Y., Y.-H.Z. and H.-J.G. synthesised and characterised the samples. E.F.C., B.M. and T.P.D. performed DFT calculations. G.H., L.P., E.F.C., T.P.D. and R.H. wrote the manuscript with comments from all the authors.

## Funding

## Competing interests

The authors declare no competing interests.

 
