## [Peer Review File · Nature Communications]

REVIEWER COMMENTS

Reviewer #1 (Remarks to the Author):

The charge-density wave (CDW) and superconductivity in kagome metal (K,Rb,Cs)V₃Sb₅ has stimulated a lot of studies. In this paper, the authors used polarization- and temperature-dependent Raman scattering to investigate the formation mechanism of the CDW in CsV₃Sb₅. The linewidths of all A_{1g} and E_{2g} phonon lines including those emerging below T_{CDW} reveal strong phonon-phonon coupling. The A_{1g} amplitude mode (AM) displays an asymmetric Fano-type lineshape, suggestive of strong electron-phonon coupling. They conclude that the co-operation of strong electron-phonon and phonon-phonon coupling may be more likely a route to the CDW transition in CsV₃Sb₅.

I think their results are interesting, but the significance is limited. There were several earlier Raman scattering studies and many other experiments to investigate the formation mechanism of the CDW in CsV₃Sb₅. I would not recommend the publication of this paper in Nature Communications.

Some minor comments:

(1) In the Samples part of Methods: "600 C", "1000 C", "2 C/h" should be "600 °C", "1000 °C", "2 °C/h".

(2) Figure 1b in the Supplementary Materials: It is a CDW transition, not a superconducting transition. Why the authors used such a small field (1 Oe, too small to me) to measure the magnetization? In Ref. 5, the magnetization was measured in 1 T (10000 Oe). In Ref. 5, there is no diamagnetic signal. Why there is diamagnetic signal in this study (which is usually observed in superconductors)? Why the author calculated $4\pi x$ (the y axis of Figure 1b in the Supplementary Materials), which is usually the superconducting volume fraction for superconductors?

Reviewer #2 (Remarks to the Author):

The authors report polarization- and temperature-resolved Raman scattering data on the kagome metal CsV₃Sb₅ with a focus on the formation mechanism of the charge-density-wave (CDW) phase. They find
- an exceptionally large CDW gap, beyond mean-field theory

- phonon anomalies in terms of frequency and linewidth at T_{CDW}
- electronic Raman scattering which, together with DFT calculations, favor an inverse star of David CDW structure rather than a star of David CDW
- an exotic asymmetric Fano lineshape of a CDW amplitude mode, indicative of substantial el-ph coupling

In light of these results, the authors emphasize the relevance of both strong phonon-phonon coupling and strong electron-phonon coupling on the formation of the CDW phase in CsV₃Sb₅.

The present study builds upon previous investigations of vanadium-based kagome metals, among them works highlighting

- strong electron-phonon coupling (Nat. Commun. 13, 3461)
- phonon anomalies at the transition temperature and the emergence of numerous zone-folded phonons and CDW amplitude modes (Nat. Commun. 13, 3461; Phys. Rev. B 105, 155106; Phys. Rev. Res. 4, 023215)
- a large CDW gap, beyond mean-field theory (Phys. Rev. B 104, L161112; Nat. Commun. 13, 273)

While the study appears technically sound and both data and analysis of high quality appropriate for an outlet like Nature Communications, I am hesitant to recommend the paper for publication in its current form. In light of the above-mentioned existing studies, the true novelty of this study evades me. For example, to what extent does it advance our current understanding of CDW formation in these compounds? To the best of my knowledge, this may be the first report of the momentum dependence of the CDW gap in CsV₃Sb₅ captured with Raman spectroscopy, as well as of the asymmetric lineshape of the amplitude mode. Perhaps the authors can more clearly distinguish their work from previous (Raman- as well as related) studies and contextualize the importance of their new findings within the broader field of 2D CDW materials.

There are a few minor technical issues, inconsistencies, and suggestions:

- the thermally-induced spectral redistribution of the electronic Raman continuum shown in the small insets in Fig. 2 is not that clear. Perhaps it would be better to plot the difference between room temperature spectra and low temperature spectra.
- the authors emphasize a "precursor" to the A_{1g} phonon anomaly with an onset of about 20 K above T_{CDW} . This claim does not seem to be supported by the presented data, or at least it is not visible to me in the current presentation. In addition, its significance is not further elaborated on throughout the manuscript.

- on the other hand, the A_{1g} phonon shows an abrupt hardening at its lowest temperature, 9 K (see Fig. 3a). This is an interesting observation if it could be linked to a precursor of the incipient superconducting phase. This hardening is however not reflected in the extracted phonon frequency plot of Fig. 3c.

- In the text the authors claim that "Both the A_{1g} and the E_{2g} phonons show significant renormalization effects at T_{CDW}". Yet, a few sentences later they state "The energy of the E_{2g} line does not exhibit significant changes across T_{CDW}".

- The statement "We ... find the Fano line to reproduce the data significantly better" is somewhat misleading, when in fact it only yields a marginally better fit than two Lorentzians. Concerning this issue, in several previous Raman studies a weak low-energy shoulder to the amplitude mode has been clearly identified, which would rationalize the fitting with two individual Lorentzians. However, this shoulder appears to be of different symmetry than the amplitude mode (as also shown in Suppl. Fig. 4). Given the otherwise clean polarization-resolved spectra without any obvious unwanted leakage presented in the current work, I tend to trust the authors on their interpretation of the amplitude mode as a single Fano-shaped line.

Reviewer #3 (Remarks to the Author):

The manuscript focuses on studying the origin of CDW transition in the Kagome metal CsV₃Sb₅ by performing the polarization- and temperature-dependent Raman scattering measurements. In addition, through the combination of experiment, DFT and mean-field theory, the authors conclude that the strong coupling scenario, including anharmonic phonon-phonon and electron-phonon coupling are more likely to result in the CDW transition. Overall, the work did careful measurements and data analyses. However, for it to be published on "Nature Communications", some unclear points to me still need to be clarified. The detailed comments are listed as follows.

(1) I am a bit confused regarding the DFT calculation on the SoD and iSoD distorted structures. What is the main message the authors want to deliver here? If the main message to deliver is that DFT results favour iSoD structure, then referring to the conclusion, should one, for example, expect different ph-ph/el-ph couplings in SoD and iSoD?

(2) The authors mentioned in Page 4, Line 104 that the mismatch in energy between DFT and experiment maybe considered as a result of the renormalization factor? Does the authors mean the band renormalization due to strong correlation effect? Could the authors please elaborate on this statement?

(3) Regarding Page 4, Line 111, I humbly ask, how I should understand the correlation between the discontinuity and the frequency hardening? It could be possible that I misunderstood. The sentence gives me a feeling that the discontinuity and the frequency hardening, replicated by DFT, are somehow related to each other.

(4) A small comment on the caption of Fig.2 (c), the color code seems to be inconsistent with the figure itself.

(5) Regarding the discussion in Page 7 about the phenomenon demonstrated in Fig.5 (a), the authors presumed that the decrease of asymmetry parameter towards zero temperature is a result of CDW gap opening. As I can observe from Fig.5 (a), the shape of the A_{1g} AM shows quite obvious temperature dependence even under T_{CDW}. How should one understand such temperature dependence at low temperature, because I assume that the gap opening is already there below T_{CDW}. In addition, the fitted $\lambda_{\text{ph-ph}}$ as shown in Fig. 3 does not reveal temperature-dependence below T_{CDW}. How about el-ph coupling term? The authors should attempt to give a more in-depth analysis on this phenomenon based on my humble opinion.

**AUTHORS' REPLY TO THE COMMENTS OF THE REVIEWERS ON MS #
NCOMMS-23-39419-T/HE**

We read the remarks of the reviewers very carefully since we realized that we did not appropriately explain important points. We are particularly grateful to reviewer #2 who, in spite of the insufficient descriptions, found out those observations which reach beyond previous work, including the anisotropy in the electronic spectra and the “exotic asymmetric Fano lineshape” of the AM. Reviewer #2 suggested also a couple changes which paved the way towards new insights such as using difference spectra for the temperature dependence of the electronic response (Fig. 2 of the manuscript), elaborating on the precursor of the A_{1g} phonon (Fig. 3) and clarifying details of the Fano shape of the AM (Fig. 5). The difference spectra unveiled the so far hidden gap-like structures in the low-energy Raman spectra mirroring the small gap in the STS and ARPES. The discussion of the precursor motivated us to include data for the volume expansion and compare them with the phonon frequencies. We hope that the suggested aspects helped to improve the manuscript substantially. The price we had to pay is a complete overhaul of the manuscript which manifests itself in many strike-outs and revisions. We describe the revisions in detail below but decided to replace occasionally entire paragraphs as shown in the marked-up manuscript instead of doing that word by word. The difference spectra (insets of Fig. 2) and the inclusion of the thermal expansion (Fig. 3) are new.

Report of Reviewer #1 – NCOMMS-23-39419-T/He

Reviewer #1: *...I think their results are interesting, but the significance is limited. There were several earlier Raman scattering studies and many other experiments to investigate the formation mechanism of the CDW in CsV_3Sb_5 . I would not recommend the publication of this paper in Nature Communications.*

We reply: We understand the reviewer’s conclusion in the light of the introductory statement. We augmented the manuscript in order to better highlight the novel observations.

All the previous publications of Raman results focus on phonons and more or less agree

on the interpretation. But even in the case of the phonons precursor effects close to T_{CDW} are neither observed nor mentioned. In the context of the AMs the analyses become more controversial and are finally not conclusive. The asymmetry of the AM either escaped attention or was interpreted in a different fashion proposing a superposition of two lines. Although we performed measurements with three laser energies (647 or 676 nm were not available) we could not find any indication of a double peak structure. Rather, the A_{1g} AMs measured at 476, 514 and 575 nm have the same shape independent of excitation. Admittedly, their resonance energy depends slightly on excitation for reasons we do not know. Thus we cannot exclude that a resonance effect in the red (647.1 nm) used in Ref. [1] produces two separate peaks, at least at low temperature. Thus, in addition to the resonance study, we measured the temperature dependence of the A_{1g} AM in detail and compared the analyses in terms of a Fano line and of two Lorentzians as shown in Supplementary Material H, Fig. 8. At any temperature the residuum R^2 for the two-line analysis is smaller than for the Fano analysis and changes non-monotonously as do the line energies and widths. In contrast, residuum, energy, and width vary monotonously in the case of a Fano line. Thus the support for the Fano hypothesis is robust, and this seems to be a first-time observation in a CDW system with the related and described implications such as enhanced electron-phonon coupling. In addition to the observation, we find a strong temperature dependence of the asymmetry parameter $1/|q|$ which peaks between 60 and 70 K just where the $2 \times 2 \times 2$ order is found to switch to a $2 \times 2 \times 4$ stacking [2].

We also studied the high-energy part of the spectra far above the phonon energies. We find a symmetry dependent redistribution of the response which can consistently be described by DFT simulations (admittedly without selection rules for reasons described in the updated manuscript). This agreement clarifies the order in the CDW phase to be tri-hexagonal or an inverse Star of David which cannot be settled with x-ray diffraction [2] or by symmetry arguments in Raman (see manuscript). As suggested by Reviewer #2 we now plot difference spectra in the insets of Fig. 2 **a** and **b**. As a results we find additional features in the 500 cm^{-1} range which resemble the small gaps also found in the single-particle methods (STS and ARPES) without exactly matching them. Wiggles in the $500\text{-}700 \text{ cm}^{-1}$ range are also seen in the optical conductivity [3] but they are too weak and did not make it into the text of the paper. The central point is that the various methods yield results which match only approximately. IR and Raman have different selection rules but measure occupied and

unoccupied states whereas ARPES is sensible only to occupied states and STS integrates over the entire band structure above or below E_F depending on the bias. This unexpected result resembles the reconstructed band structure in the CDW phase. The finding is rather different from the transition into a superconducting state. All these arguments are now spelled out in the updated manuscript.

Finally, we augmented the discussion of the phonons. We had already a very dense mesh of temperatures around T_{CDW} . Now, in addition to extracting the phonon-phonon coupling parameters from the line width, we include data for the volume expansion, which became available only recently [4], to highlight the anomalies of the energy around T_{CDW} . Upon including data from additional samples we find a significant dip in the energy of the A_{1g} phonon right above (not at) T_{CDW} which is very similar to the anomaly in MnSi [5] and indicates fluctuations in the phonon domain rather than in the electronic part. In other words, one may speculate that the phonon energy shows the so-far unobserved fluctuation precursor of the phase transition expected indirectly from the high $2\Delta/k_B T_{\text{CDW}}$ ratio. This was missed in prior work and is a key aspect of our work, highlighting a different perspective on the CDW transition in CsV_3Sb_5 .

Reviewer #1: *In the Samples part of Methods: "600 C", "1000 C", "2 C/h" should be "600 °C", "1000 °C", "2 °C/h".*

We reply: Fixed.

Reviewer #1: *Figure 1b in the Supplementary Materials: It is a CDW transition, not a superconducting transition. Why the authors used such a small field (1 Oe, too small to me) to measure the magnetization? In Ref. 5, the magnetization was measured in 1 T (10000 Oe). In Ref. 5, there is no diamagnetic signal. Why there is diamagnetic signal in this study (which is usually observed in superconductors)? Why the author calculated $4\pi\chi$ (the y axis of Figure 1b in the Supplementary Materials), which is usually the superconducting volume fraction for superconductors?*

We reply: We previously used data measured with the magnetic field parallel to the ab -plane of the sample, where the magnetization is extremely small and even exhibits diamagnetism sometimes. To avoid confusion, we used data obtained with a field (induction) of 1 T perpendicular to the planes and replaced Figure 1b in the Supplementary Materials

with Fig. R1. The units on the ordinate are now the usual ones.

FIG. R1. Temperature dependence of magnetization. The field is perpendicular to the ab -plane of the sample. The CDW transition occurs at 95 K.

Report of Reviewer #2 – NCOMMS-23-39419-T/He

Reviewer #2: *...While the study appears technically sound and both data and analysis of high quality appropriate for an outlet like Nature Communications, I am hesitant to recommend the paper for publication in its current form. In light of the above-mentioned existing studies, the true novelty of this study evades me. For example, to what extent does it advance our current understanding of CDW formation in these compounds? To the best of my knowledge, this may be the first report of the momentum dependence of the CDW gap in CsV_3Sb_5 captured with Raman spectroscopy, as well as of the asymmetric lineshape of the amplitude mode. Perhaps the authors can more clearly distinguish their work from previous (Raman- as well as related) studies and contextualize the importance of their new findings within the broader field of 2D CDW materials.*

We reply: We are very grateful to Reviewer #2 for clearly spelling out the issue. Fortunately, Reviewer #2 captured the crucial point which we certainly did not highlight and explain sufficiently in the manuscript.

We made every effort to fix the problem. We kindly ask Reviewer #2 to refer to the first Reply to Reviewer #1 who addressed the same problem.

We augmented the description of the context concerning the electronic anisotropy in view of earlier optical results and single-particle spectroscopies. New insight became possible after plotting the difference spectra in the insets of Fig. 2 (see next comment). The difference spectra show additional features which were hidden in the raw data: (1) there are symmetry dependent low-energy features in the range $400\text{-}700\text{ cm}^{-1}$ which are close to the low-energy gap in ARPES and STS if a factor of two for Δ vs. 2Δ is taken into account. The Raman features at above 1500 cm^{-1} appear at higher energies than that in the IR spectra [3] and in ARPES. We do not yet have a quantitative explanation for reasons spelled out in the new manuscript (problems with the gauge invariance for interband transitions [6]) but it is clear on the other hand that the unoccupied states in a CDW system are not symmetric to the occupied states as in a superconductor. As a matter of fact, we observe two distinct energy scales and show that they are linked directly to the iSoD (tri-hexagonal) distortion of the lattice in the CDW state.

Reviewer #2: *The thermally-induced spectral redistribution of the electronic Raman continuum shown in the small insets in Fig. 2 is not that clear. Perhaps it would be better to plot the difference between room temperature spectra and low temperature spectra.*

We reply: To enhance the features induced by the redistribution of the Raman spectra below T_{CDW} , we subtracted the spectra measured at 124 K from those at lower temperature. In the insets of Fig. 2 **a** and **b** we show now the difference spectra which, to our own surprise, unveil additional features: There are structures at 600 cm^{-1} (75 meV) and 450 cm^{-1} (56 meV) for A_{1g} and E_{2g} symmetry, respectively, and the high-energy part of the E_{2g} spectra consists of two distinct temperature independent structures at 2100 ± 200 and $3000 \pm 200 \text{ cm}^{-1}$. These new observations were facilitated by the request of Reviewer #2 “to plot the difference between room temperature spectra and low temperature spectra”. Since nothing interesting happens above T_{CDW} we took the freedom to use the 124-K spectra as a basis.

Reviewer #2: *The authors emphasize a “precursor” to the A_{1g} phonon anomaly with an onset of about 20 K above T_{CDW} . This claim does not seem to be supported by the presented data, or at least it is not visible to me in the current presentation. In addition, its significance is not further elaborated on throughout the manuscript.*

We reply: We appreciate the Referee pointing this out, which has allowed us to more clearly define and support our claim on the precursor. We augmented the figure, include now data for the volume expansion and apply Grüneisen theory. The volume expansion ($\alpha_V(T)$ is shown in Fig. 3 **h**) explains the E_{2g} data but not those in A_{1g} symmetry, where the increase of the phonon energy by approximately 1 cm^{-1} below T_{CDW} originates from the new positions of the V atoms.

In order to visualize the precursor effect we now show the A_{1g} energies on an expanded scale in Fig. 3 **a** of the manuscript for three different samples. We show the results also here in Fig. R2. Beyond minor variations between the samples we consistently observe a dip in the energy between 101 and 102 K. This precursor is similar to the one observed for MnSi just above the magnetic transition [5]. When we include the Grüneisen result from the thermal expansion it is not justified to speak about a 20-K range. We remove that from

the text and discuss the dip. We argue that the dip is another indication of enhanced el-ph coupling entailing unharmonic ph-ph decay. Currently, there is no access to a theoretical description.

FIG. R2. A_{1g} phonon energy near T_{CDW} as a function of temperature. S1, S2 and S3 denote different samples. While there are differences between the samples in the range $\pm 0.5 \text{ cm}^{-1}$ and also in the temperature dependence around T_{CDW} the dip between 101 and 102 cm^{-1} can be considered significant. Similarly, the hardening below T_{CDW} is significant whereas an abrupt change at T_{CDW} needs to be settled by further experiments. The precursor is limited to a range of 10 K.

Reviewer #2: *on the other hand, the A_{1g} phonon shows an abrupt hardening at its lowest temperature, 9 K (see Fig. 3a). This is an interesting observation if it could be linked to a precursor of the incipient superconducting phase. This hardening is however not reflected in the extracted phonon frequency plot of Fig. 3c.*

We reply: We appreciate that the Reviewer revealed this observation from the data. Previously, we used data from Sample #3 (Fig. R3 h) for the Raman spectra, while the phonon energy versus temperature plot is derived from the data of Sample #1 (Fig. R3 c).

This inconsistency has led to confusion. Admittedly, we have not reached any agreement regarding the precursor of the incipient superconducting phase due to variations among different samples. To rectify this, we have replaced the Raman spectra in the A_{1g} symmetry with the data taken from Sample #1 in Fig. 3 of the main text.

FIG. R3. Raman spectra in the A_{1g} symmetry and the corresponding phonon energy and linewidths from the Voigt fits. The data are taken from three different samples.

Reviewer #2: *In the text the authors claim that “Both the A_{1g} and the E_{2g} phonons show significant renormalization effects at T_{CDW} ”. Yet, a few sentences later they state*

"The energy of the E_{2g} line does not exhibit significant changes across T_{CDW} ".

We reply: Indeed, the energy of the E_{2g} phonon does not exhibit significant changes across T_{CDW} . The E_{2g} phonon has only a kink in the linewidth at T_{CDW} . We have included the new subsection "Persistent phonons" in the manuscript describing in some detail the various effects.

Reviewer #2: *The statement "We ... find the Fano line to reproduce the data significantly better" is somewhat misleading, when in fact it only yields a marginally better fit than two Lorentzians. Concerning this issue, in several previous Raman studies a weak low-energy shoulder to the amplitude mode has been clearly identified, which would rationalize the fitting with two individual Lorentzians. However, this shoulder appears to be of different symmetry than the amplitude mode (as also shown in Suppl. Fig. 4). Given the otherwise clean polarization-resolved spectra without any obvious unwanted leakage presented in the current work, I tend to trust the authors on their interpretation of the amplitude mode as a single Fano-shaped line.*

We reply: We agree with the Reviewer that the residuum R^2 at a single temperature is not overly convincing. Since we agree also that the Fano shape is kind of unique we discuss the issue more comprehensively. In other words, the paragraph is completely rewritten and refers explicitly to Supporting Material H Fig. 8. There we show R^2 for all temperatures studied and find that the analysis in terms of a Fano line yields monotonous and smooth temperature dependences of both resonance energy and line width.

Furthermore, our Raman spectra never exhibit a clear double-peak structure for the A_{1g} amplitude mode across the entire temperature range below T_{CDW} . We wish to emphasize that our signal-to-noise ratio is very high for the A_{1g} amplitude mode. As mentioned by the referee, the low-energy shoulder observed in previous Raman studies may be attributed to leakage. Therefore, we prefer to explain the observed phenomenon as a Fano resonance rather than a a superposition of two Lorentzian lines.

Report of Reviewer #3 – NCOMMS-23-39419-T/He

We thank the Referee for a careful reading of our manuscript. The Referee has raised a number of points largely connected to our theory work. We have used these comments to help clarify, amend, and correct our discussion in the revised manuscript. Our point-by-point discussion to the Referee’s comments are given below.

Reviewer #3: *I am a bit confused regarding the DFT calculation on the SoD and iSoD distorted structures. What is the main message the authors want to deliver here? If the main message to deliver is that DFT results favour iSoD structure, then referring to the conclusion, should one, for example, expect different ph-ph/el-ph couplings in SoD and iSoD?*

We reply: We only focus on the gap opening and find that the iSoD distortion entails a stronger reduction of the Raman response below the intersection point than the SoD distortion thus favoring the iSoD pattern. Calculating differences in the el-ph coupling for the various distortion patterns is currently not reliable. The study of ph-ph interaction is even more difficult and thus, while one might expect different phonon-phonon couplings, it is beyond the scope of this work.

Reviewer #3: *The authors mentioned in Page 4, Line 104 that the mismatch in energy between DFT and experiment maybe considered as a result of the renormalization factor? Does the authors mean the band renormalization due to strong correlation effect? Could the authors please elaborate on this statement?*

We reply: We thank the Referee for raising this point and allowing us to clarify what was meant. The mismatch in the electronic energies between DFT and experimentally observed ARPES dispersions is a result of band renormalization due to strong correlation effects which are expected in these materials and were shown to exist by IR, e.g. [7]. This reference and an extended explanation is now included at the end of the subsection “Electronic continuum”.

Reviewer #3: *Regarding Page 4, Line 111, I humbly ask, how I should understand the*

correlation between the discontinuity and the frequency hardening? It could be possible that I misunderstood. The sentence gives me a feeling that the discontinuity and the frequency hardening, replicated by DFT, are somehow related to each other

We reply: We apologize for the confusion. We agree that the two phenomena are related. Collecting data from all samples we have to conclude that a discontinuous change at T_{CDW} appears only in one sample and should thus not be claimed as general. The changes at T_{CDW} are fast but probably not discontinuous. In addition, the energy renormalization effects vary from sample to sample but are by and large matched by the DFT simulations. The dip in energy at 101-102 K are a general observation.

Reviewer #3: *A small comment on the caption of Fig.2 (c), the color code seems to be inconsistent with the figure itself.*

We reply: We have corrected that in the manuscript.

Reviewer #3: *Regarding the discussion in Page 7 about the phenomenon demonstrated in Fig.5 (a), the authors presumed that the decrease of asymmetry parameter towards zero temperature is a result of CDW gap opening. As I can observe from Fig.5 (a), the shape of the A1g AM shows quite obvious temperature dependence even under T_{CDW} . How should one understand such temperature dependence at low temperature, because I assume that the gap opening is already there below T_{CDW} . In addition, the fitted $\lambda_{\text{ph-ph}}$ as shown in Fig. 3 does not reveal temperature-dependence below T_{CDW} . How about el-ph coupling term? The authors should attempt to give a more in-depth analysis on this phenomenon based on my humble opinion.*

We reply: The AM appears only below T_{CDW} . Close to T_{CDW} , the width is almost as large as the energy. We do not believe that the width is related to strong ph-ph coupling. Unfortunately, this aspect is not sufficiently clear in item (iii) towards the end of the first paragraph on amplitude modes. What we wanted to say is that strong ph-ph coupling may influence the temperature dependence of the AM according to Ref. [8]. As a matter of fact, we cannot extract the ph-ph coupling from the AM but only from phonons having a regular temperature dependence. We reformulated this point and emphasize here the importance of el-ph coupling as intended.

-
- [1] S. Wu, B. R. Ortiz, H. Tan, S. D. Wilson, B. Yan, T. Birol, and G. Blumberg, Charge density wave order in the kagome metal AV_3Sb_5 ($A = Cs, Rb, K$), Phys. Rev. B **105**, 155106 (2022).
- [2] Q. Stahl, D. Chen, T. Ritschel, C. Shekhar, E. Sadrollahi, M. C. Rahn, O. Ivashko, M. v. Zimmermann, C. Felser, and J. Geck, Temperature-driven reorganization of electronic order in CsV_3Sb_5 , Phys. Rev. B **105**, 195136 (2022).
- [3] E. Uykur, B. R. Ortiz, O. Iakutkina, M. Wenzel, S. D. Wilson, M. Dressel, and A. A. Tsirlin, Low-energy optical properties of the nonmagnetic kagome metal CsV_3Sb_5 , Phys. Rev. B **104**, 045130 (2021).
- [4] M. Frachet, L. Wang, W. Xia, Y. Guo, M. He, N. Maraytta, R. Heid, A.-A. Haghighirad, M. Merz, C. Meingast, and F. Hardy, Colossal c-axis response and lack of rotational symmetry breaking within the kagome plane of the CsV_3Sb_5 superconductor, arXiv:2310.06102 (2023).
- [5] H.-M. Eiter, P. Jaschke, R. Hackl, A. Bauer, M. Gangl, and C. Pfleiderer, Raman study of the temperature and magnetic-field dependence of the electronic and lattice properties of $MnSi$, Phys. Rev. B **90**, 024411 (2014).
- [6] M. Schüler, J. A. Marks, Y. Murakami, C. Jia, and T. P. Devereaux, Gauge invariance of light-matter interactions in first-principle tight-binding models, Phys. Rev. B **103**, 155409 (2021).
- [7] E. Uykur, B. R. Ortiz, S. D. Wilson, M. Dressel, and A. A. Tsirlin, Optical detection of the density-wave instability in the kagome metal KV_3Sb_5 , npj Quantum Materials **7**, 16 (2022).
- [8] C. M. Varma and A. L. Simons, Strong-coupling theory of charge-density-wave transitions, Phys. Rev. Lett. **51**, 138 (1983).

REVIEWER COMMENTS

Reviewer #1 (Remarks to the Author):

The authors replied all the comments from three reviewers carefully, and made substantial revision of the manuscript to highlight the novelty of their study.

Now I think it is fine for the publication of this paper in Nature Communications.

Reviewer #2 (Remarks to the Author):

I appreciate the authors' substantial revisions to their manuscript, including modifications of their figures, and expanding their discussion. I believe that these changes greatly help to strengthen their claims. I have a few minor follow-up comments, but in principle I would support publication of the revised manuscript in Nat. Commun.

1. I am happy to see that the authors included plots of difference spectra in their Figures 2a and 2b, which helps the readers to appreciate the temperature evolution in detail. It would be very interesting to see the spectral weight (the integrated intensity) of these difference spectra plotted against the temperature. Particularly, the high energy part of the electronic continuum measured in the E_{2g} channel seems to increase abruptly between 44 K and 25 K. Could this hint to an intermediate transition between T_{CDW} and T_c, as reported by several groups?

2. Fig. 3: Error bars to extracted phonon parameters should be included.

3. The observed "precursor phase", deduced from deviations from the Grüneisen behavior of the A_{1g} phonon energy, may be related to recently reported experimental evidence for the emergence of nematicity above T_{CDW} (see <https://arxiv.org/abs/2309.16985>). As such, these Raman results would serve as important additional, independent experimental fingerprints.

4. Language: A few statements that sound ambiguous: "line widths may display a weak peak right at the transition" -- perhaps the authors want to say the line widths slightly increase at the transition temperature; energy of the phonon "goes in the wrong direction" -- what is a wrong direction? Contrary

to a softening as expected from the volume expansion data below T_{CDW} , the phonon frequency hardens. "The continuously increasing spectra..." -- What are increasing spectra? Perhaps the continuously increasing spectral weight with increasing Raman shift? Or with decreasing temperature?

Reviewer #3 (Remarks to the Author):

I believe that the manuscript is now qualified to be published on Nature Communications.

One small error I pointed out is the orange diamond symbol in Fig. 4 a and b, which is displayed as "à" for some reason.

**AUTHORS' REPLY TO THE COMMENTS OF THE REVIEWERS ON MS #
NCOMMS-23-39419-A/HE**

We thank all the Reviewers for critically reading our manuscript and their helpful remarks. In response to the suggestive comments from the Reviewer #2, we added error bars in Fig. 3. In addition, we worked on the language and fixed the statement on the comparison between phonon energy and Grüneisen predictions in lines 198 through 200 of the marked-up manuscript. Finally, we replaced "The continuously increasing..." by "The redistribution of the spectral weight below T_{CDW} exhibiting..." which is hopefully more clear.

Report of Reviewer #1 – NCOMMS-23-39419-A/He

Reviewer #1: *The authors replied all the comments from three reviewers carefully, and made substantial revision of the manuscript to highlight the novelty of their study. Now I think it is fine for the publication of this paper in Nature Communications.*

We reply: We are very grateful to the reviewer for the positive recommendation.

Report of Reviewer #2 – NCOMMS-23-39419-A/He

Reviewer #2: *I appreciate the authors' substantial revisions to their manuscript, including modifications of their figures, and expanding their discussion. I believe that these changes greatly help to strengthen their claims. I have a few minor follow-up comments, but in principle I would support publication of the revised manuscript in Nat. Commun.*

We reply: We sincerely appreciate the reviewer for the thoughtful suggestions and the decision to accept our manuscript for publication. Your insights and feedback have greatly contributed to enhancing the quality and clarity of our work.

Reviewer #2: *I am happy to see that the authors included plots of difference spectra in their Figures 2a and 2b, which helps the readers to appreciate the temperature evolution in detail. It would be very interesting to see the spectral weight (the integrated intensity) of these difference spectra plotted against the temperature. Particularly, the high energy part of the electronic continuum measured in the E_{2g} channel seems to increase abruptly between 44 K and 25 K. Could this hint to an intermediate transition between T_{CDW} and T_c , as reported by several groups?*

We reply: As seen in Fig. R1, the difference spectra in the E_{2g} channel integrated between the intersection points and 3600 cm^{-1} exhibit a sharper increase between 44 K and 25 K as opposed to the A_{1g} spectra. Since we calculated Raman spectra only for 2D distortions (SoD and iSoD) we do not have access to the effect of other distortions [1–4] on the Raman spectra. In addition, we cannot quantitatively include the selection rules nor do we expect general sum rules for Raman spectra (see also Ref. 35 of the manuscript).

Reviewer #2: *Fig. 3: Error bars to extracted phonon parameters should be included.*

We reply: Error bars are included.

Reviewer #2: *The observed "precursor phase", deduced from deviations from the Grüneisen behavior of the A_{1g} phonon energy, may be related to recently reported experimental evidence for the emergence of nematicity above T_{CDW} (see <https://arxiv.org/abs/2309.16985>)*

FIG. R1. Integrated difference spectral weight from 1500 cm^{-1} to 3600 cm^{-1} in A_{1g} and E_{2g} symmetries.

As such, these Raman results would serve as important additional, independent experimental fingerprints.

We reply: The nematicity transition reported by T. Asaba *et al.* occurs at $T^*=130 \text{ K}$ and is electronic, with possibly odd parity. It is unlikely to be observable by Raman scattering in a material having inversion symmetry where odd and even excitations can be distinguished. Nematicity with odd parity may couple to IR-active phonons but not to Raman-active phonons with even parity. The "precursor" we have observed in the A_{1g} phonon energy indicates more likely a coupling to the CDW order parameter but there is no theoretical support for this speculation.

Reviewer #2: *Language: A few statements that sound ambiguous: "line widths may display a weak peak right at the transition" perhaps the authors want to say the line widths slightly increase at the transition temperature; energy of the phonon "goes in the wrong direction" what is a wrong direction? Contrary to a softening as expected from the volume expansion data below T_{CDW} , the phonon frequency hardens. "The continuously increasing*

spectra...” What are increasing spectra? Perhaps the continuously increasing spectral weight with increasing Raman shift? Or with decreasing temperature?

We reply: We appreciate the Reviewer for pointing out the inaccurate expressions. The corresponding sentences now have been modified in the manuscript.

Report of Reviewer #3 – NCOMMS-23-39419-A/He

Reviewer #3: *I believe that the manuscript is now qualified to be published on Nature Communications. One small error I pointed out is the orange diamond symbol in Fig. 4 a and b, which is displayed as "à" for some reason.*

We reply: We thank the Referee for his positive comments and the recommendation for publication. Figure 4 has been updated to remove the small error.

-
- [1] H. Zhao, H. Li, B. R. Ortiz, S. M. L. Teicher, T. Park, M. Ye, Z. Wang, L. Balents, S. D. Wilson, and I. Zeljkovic, Cascade of correlated electron states in the kagome superconductor CsV_3Sb_5 , Nature **599**, 216 (2021).
 - [2] Z. Liang, X. Hou, F. Zhang, W. Ma, P. Wu, Z. Zhang, F. Yu, J.-J. Ying, K. Jiang, L. Shan, Z. Wang, and X.-H. Chen, Three-dimensional charge density wave and surface-dependent vortex-core states in a kagome superconductor CsV_3Sb_5 , Phys. Rev. X **11**, 031026 (2021).
 - [3] Q. Stahl, D. Chen, T. Ritschel, C. Shekhar, E. Sadrollahi, M. C. Rahn, O. Ivashko, M. v. Zimmermann, C. Felser, and J. Geck, Temperature-driven reorganization of electronic order in CsV_3Sb_5 , Phys. Rev. B **105**, 195136 (2022).
 - [4] L. Nie, K. Sun, W. Ma, D. Song, L. Zheng, Z. Liang, P. Wu, F. Yu, J. Li, M. Shan, D. Zhao, S. Li, B. Kang, Z. Wu, Y. Zhou, K. Liu, Z. Xiang, J. Ying, Z. Wang, T. Wu, and X. Chen, Charge-density-wave-driven electronic nematicity in a kagome superconductor, Nature **604**, 59 (2022).

REVIEWERS' COMMENTS

Reviewer #2 (Remarks to the Author):

The authors have thoroughly addressed all raised issues in their revision. I am therefore happy to recommend the publication of their manuscript in Nat. Commun.

**AUTHORS' REPLY TO THE COMMENTS OF THE REVIEWERS ON MS #
NCOMMS-23-39419-B/HE**

Report of Reviewer #2 – NCOMMS-23-39419-B/He

Reviewer #2: *The authors have thoroughly addressed all raised issues in their revision. I am therefore happy to recommend the publication of their manuscript in Nat. Commun.*

We reply: We thank the referee again for the positive recommendation.